# Acute and Persistent Postoperative Functional Decline in Children with Severe Neurological Impairment: A Qualitative, Exploratory Study

**DOI:** 10.3390/children11030319

**Published:** 2024-03-07

**Authors:** Liisa Holsti, Sarah England, Mackenzie Gibson, Bethany McWilliams, Anne-Mette Hermansen, Harold Siden

**Affiliations:** 1Department of Occupational Science and Occupational Therapy, University of British Columbia, Vancouver, BC V6T 2B5, Canada; sengland@providencehealth.bc.ca (S.E.); mackenzie.gibson@islandhealth.ca (M.G.);; 2Department of Pediatrics, University of British Columbia, Vancouver, BC V6H 3V4, Canada; ahermansen@bcchr.ca (A.-M.H.); hsiden@bcchr.ca (H.S.); 3BC Children’s Hospital Research Institute, Vancouver, BC V5Z 4H4, Canada

**Keywords:** pediatrics, caregivers, severe neurological impairment, complex care, surgery, function

## Abstract

Background: Children with severe neurologic impairment (SNI) regularly require major surgery to manage their underlying conditions. Anecdotal evidence suggests that children with SNI experience unexpected and persistent postoperative functional changes long after the postoperative recovery period; however, evidence from the perspective of caregivers is limited. The purpose of the study was to explore the functional postoperative recovery process for children with SNI. Methods: Eligible participants were English-speaking caregivers of children with SNI between 6 months and 17 years who were nonverbal, Gross Motor Function Classification Scale level IV/V, and who had surgery/procedure requiring general anesthetic at a tertiary children’s hospital between 2012 and 2022. Demographic and basic health information were collected via surveys and corroborated by a review of the child’s electronic health record. Semi-structured interviews were conducted and a thematic content analysis was used to formulate results. Results: Data from 12 primary caregiver interviews revealed four main themes: (1) functional changes and complications in the child; (2) feeling unprepared; (3) perioperative support; and (4) changes to caregiver roles. Conclusions: Postoperative functional decline in children with SNI was prevalent in our sample. Providing pre-operative information to families to describe this phenomenon should be a regular part of family-informed care.

## 1. Introduction

Children with severe neurologic impairment (SNI) have significant neurological disorders that result in developmental impairments influencing intellectual function, tone/motor control/mobility (cerebral palsy), and communication, often with impacts on other organ systems [1]. These children often require surgical interventions, such as dental surgery and correction of deformities in the hips and spine, to manage their underlying conditions [2,3,4]. Anecdotal evidence suggests that surgical interventions may result in unanticipated acute onset and persistent postoperative developmental impairments [5]. Evidence exists for peri-operative decline in functional outcomes; however, it is predominantly reported in the frail elderly (e.g., [6,7,8,9]), and the methods used are typically quantitative. Historically, anesthesia exposure combined with pre-operative frailty has been proposed as the main cause of adverse functional changes, yet the effects in the frail elderly last far beyond the metabolic clearance of the anesthetics [7,10,11]. One study has shown that caregivers for children with low levels of function have more concerns regarding delayed recovery, immediate postoperative pain, self-care and postoperative rehabilitation than caregivers of children with better function [12]. Research specifically examining persistent postoperative functional decline in children with SNI has not been reported particularly from the caregiver perspective. Thus, the purpose of this study was to understand the experiences of children with SNI following major and minor surgical interventions as reported by their primary caregivers. 

## 2. Materials and Methods

Using convenience sampling, participants were recruited from a pediatric palliative care center and an orthopedic surgical clinic at a tertiary children’s hospital. The study protocol was approved by the Children’s and Women’s Research Ethics Board of the University of British Columbia (H16-02206). Those included were caregiver(s) of a child between 6 months and 17 years whose child’s condition was consistent with the description of SNI. The child had limited or no communication (Communication Function Classification System [CFCS] level IV or V), and a Gross Motor Function Classification Scale (GMFCS) level IV or V, and had undergone either a major or minor invasive procedure requiring general anesthesia [13,14,15,16,17,18,19,20]. Examples of the former included orthopedic surgical management of musculoskeletal pathology in the hips or spine, while examples of the latter included placement of a gastrostomy. Procedures were performed at a tertiary children’s hospital between January 2012 and August 2022. To validate the inclusion criteria, each caregiver also answered questions about their child’s mobility and communication to confirm their GMFCS and CFCS classification. Those excluded were caregivers who were non-English-speaking and caregiver(s) of a child with a GMFCS level of less than IV. 

All 12 semi-structured interviews were conducted via phone or Zoom™ 5.16.10 by two researchers and lasted 30–60 min. Interviews were recorded, transcribed verbatim, and saved on a secure server with password protection enabled. Pseudonyms were used for participants and all location or provider names deidentified. Participants received gift cards as a token of appreciation for their participation. Interview questions focused on caregivers’ experience and understanding of their child’s surgery/surgeries and the postoperative period for however long caregivers reported seeing continued changes to their child’s function. The interviewer’s questions were structured to gain a description of parents’ observations of their child’s recovery with regard to physical and cognitive function, and to discuss their child’s quality of life and the changes that occurred post-surgery. For example, the interviewers asked the following questions: “Tell me about your child’s postoperative recovery following this surgery, including complications, unanticipated events and functional recovery”, “How were you counseled about the expected postoperative course and what rehabilitation was required?” and “Were the changes seen following both major and minor procedures (e.g., gastronomy vs. spine instrumentation) or only in one or the other?”. As the interviews were semi-structured in nature, the researchers allowed for caregivers’ responses to lead to exploration of experiences or themes not envisioned in the interview guide but that were meaningful to understanding caregiver’s overall evaluation of their child’s postoperative experience of functional changes as well as their own role in supporting their child throughout the process. After 12 interviews, data sufficiency was achieved. 

The steps outlined by Braun and Clark were used to complete the thematic analysis to identify and report similarities/patterns within the data [21,22,23]. Interview notes and observations supplemented the analysis process [24]. The researchers considered their own positionality as they narrowed the views from the interview and generated themes from their content [25]. The researchers collaboratively reviewed and produced codes for one interview to ensure that they had a shared understanding of the coding process. Specifically, researchers approached coding by highlighting content of the interviews that was related to the interview questions with specific color codes [26,27,28]. Each researcher completed the process with the remaining interviews independently. Once all transcriptions were completed, the researchers collaborated to re-read, discuss, and agree upon codes for all interviews and all of the coded interview content was condensed and organized into a table. 

Four major themes were identified by combining and organizing the codes, utilizing the research objectives as a guide. A table was generated to facilitate this process of comparing codes. Texts were then re-read to ensure consistency between the initial data collected and the themes generated to help ensure trustworthiness of the analysis [27]. Additionally, two co-authors (LH, AMH) reviewed all written transcripts before reviewing data analysis findings. 

The researchers also conducted data triangulation and reflexivity by independently creating notes during interviews and reflecting together following each interview. Additionally, researchers independently answered interview questions to identify possible expectations in advance and support conscious reflection during interviews. These notes included information that may not have been present in recordings, such as the environment in which the interview took place, details about the participants (body language), notes on research partner’s cadence, as well as possible important details that might have impacted participant’s response to interview questions. 

## 3. Results

A total of 12 caregiver interviews were conducted; 11 interviews were completed with biological mothers and 1 interview was carried out with a foster mother and a paid caregiver simultaneously. The demographic and clinical information for the 12 children is summarized in Table 1. 11 children scored as V on both GMFCS and CFCS, and 1 scored as IV on both, and surgeries ranged from major hip/spine procedures to ear tube and gastrostomy tube placement.

Although interview questions focused on the temporary or persistent experiences of functional change in their children, caregivers provided much additional and rich information on the recovery experience. Therefore, four themes were identified: (1) functional changes and complications in the child; (2) feeling unprepared; (3) perioperative support; and (4) changes to caregiver roles.

### 3.1. Theme 1: Functional Changes and Complications in the Child 

All 12 interviewees noted various postoperative complications and functional decline in their children (see Table 2). The duration of recovery and intensity of decline varied and is highlighted in Table 3. Erik’s caregiver reported, like 9 other participants, that he experienced significant declines that impacted his quality of life.


*Prior to that [surgery,] he enjoyed standing, he enjoyed eating; he actually loved McDonald’s happy meals, that kind of thing. Everything after surgery, his decline in his ability to chew properly, swallow, all of that changed… He actually never really stood after that.*
(Katie)

Charlie’s changes in seizures postoperatively altered his ability to participate in daily activities.


*It’s definitely impacted things. The 10 s short seizures don’t really seem to bother him, he goes about his activities afterwards, but with the longer ones quite often he’ll need a nap afterwards. So, if that happens on the way to school he’ll be late to school, it has happened two times in class now where other students have noticed his seizures, and it has happened a couple of times where I have to come pick him up because he is just so tired after. It has also resulted in him missing out on going to a swimming pool every Monday with school because he had had a big seizure just before or earlier in the day and that is his favourite place to go!*
(Nadine)

Four caregivers noted changes in communication after surgery, in part because of their child’s decreased physical dexterity, which is required for signing or operating assistive technology.


*Prior to those two surgeries we had been using the eye gaze computer quite regularly. I don’t want to use the word efficiently because it wasn’t like we could give him this computer and he could communicate without a communication partner, but he did have pretty good use of choosing things and playing things. That’s sitting in the corner getting dust at school. We’ve had to switch to low-tech stuff we can move around in place for him instead of him moving his head around [to operate].*
(April)

Other physical and emotional effects due to complications were also described. “During that surgery, he had a lot more blood loss and his seizures increased; he started having more seizures and longer seizures. So, there were just more problems in general after that surgery” (Anne). Caregivers further described postoperative stress symptoms in their child after major invasive surgeries. For example, Katie acknowledged the functional impacts that her child’s distress caused. 


*It was determined that he was really doing quite a bit of obstructive breathing apnea and I think he was in a lot of pain; he was breath holding, that kind of thing. So, there were a lot of behaviors that kind of came from the expense of that. He was finally able to be moved to sitting in a chair for example, but he never wanted to leave his room, he never wanted to leave the kitchen area, he never wanted to go for walks, he would just literally start screaming. He was just so anxious because at that time we did have several trips to the hospital.*
(Katie)

### 3.2. Theme 2: Feeling Unprepared

Caregivers understood the surgeries their children had undergone to be necessary and had carefully considered the benefits of the surgeries as well as their timing (to the extent that it was within their control). They acknowledged, however, that their preparedness for the postoperative period and potential functional changes in their child was lacking. 


*He [Erik] had a lot done in that surgery, and in their [surgeon’s] mind it was successful, in a sense, but it was much more of a surgery than we ever anticipated, to be honest… We could only do one side [of the hip reconstruction], we couldn’t do the other side because of his really poor recovery from it… So he is left with now almost a displaced, dislocated other side that we can’t do anything about.*
(Katie)

Caregivers recalled how they did not expect things such as a loss of sensation, increased seizures, changes to communication, or for wounds to take so long to heal after the surgical procedure. Others explained how they thought their child’s comfort would improve after surgery or their feeding would normalize once things like the full-body spica casts were removed after hip surgery. Jean recalled being surprised that her son’s cough weakened after surgery. She explained how she now has to lay her son down regularly throughout the day to help clear his secretions, and noted how this has impacted his ability to participate in activities outside of the home.


*What we were thinking is that once surgery is done, he [James] is going to be so much more comfortable sitting up in the chair because then his lungs are clear and he’s not bent over, so this was kind of a surprise to me.*
(Jean)

Some families were surprised that the long recovery presented new challenges.


*It was pretty much during that time of him recovering from the hip surgery that obviously he had gotten weaker and so then the scoliosis… just took right over. So, at that point it was like, we’re still in recovery mode and then all of a sudden, we are having to think about spinal surgery.*
(Katie)

Every child’s recovery experience was unique; however, 9 out of 12 families expressed surprise at the challenges with pain management after major surgery. Katie shared: “*That was really hard for everybody, just because we didn’t expect his pain to be so bad for so long… It took probably well over a year until he was more himself.”* Caregivers acknowledged increased challenges with pain management due to their children’s nonverbal state and revealed their frustrations with how pain was managed in the hospital setting. 


*I think it’s a trust you have in your healthcare providers that they are gonna care for your child. I mean Taylor isn’t unique at [the tertiary children’*
*s hospital], I mean Taylor would be unique at [the regional general hospital] or [local community hospital] where they don’t have kids like her, but they [the tertiary children’s hospital] specialize in caring for kids like her who are nonverbal, where you’re going off physical or behavioral cues for pain. So, I think with that I just would have expected better of them.*
(Nicole)

Caregivers were often the first to identify when their children were experiencing pain and had to advocate in order for their needs to be met. 


*I talked to the surgeon and the surgeon’s like, ‘okay maybe it could be a new behavior that’s emerging from the spinal surgery’ and ‘oh these are seizures.’ And I’m letting you know they’re not seizures. This child is literally in pain, he just can’t tell us.*
(Kate)

Communication barriers between healthcare practitioners and the children often required caregivers to enlist strategies to support communication related to pain management.


*I drew a picture and asked him [Erik] to point to where his pain was coming from and he actually had a pressure sore so that was able to be discovered and treated before it got really, really bad. He still has a scar today, but it could have been much worse.*
(Katie)

Facing changes in their child’s function and feeling unprepared for them were augmented for some families by their frustration with the continuity of care and perceived lack of communication between healthcare providers throughout the postoperative recovery process. Beth explained “*I think the biggest problem is that there are so many holes after surgery, nobody talks to anybody else and so you end up leaving the hospital not sure what on earth you are going to do in a certain area.*” Beth further explained the importance of coordinated care when having numerous care providers involved, stating “*He [Tommy] has a different team for every part of his body, basically*.” Caregivers highlighted the negative consequences that result from fragmented and siloed health care.


*I find that the two [hospitals] don’t communicate very efficiently together, so that’s been challenging, especially when there’s been medicine prescribed by the hospital here [locally], but then I’m trying to get our tone management and even our seating at [the tertiary children’s hospital] notified as to what’s going on. It goes into the abyss somewhere and I think it would be a lot more helpful if they all worked together.*
(April)

With respect to sharing information between the various care providers, families noted they had to act as a bridge to ensure necessary information was communicated.


*I just keep telling everyone I don’t know who I’m supposed to ask. Everyone keeps asking what are we supposed to be doing? And so, we just kind of sit on the back burner. And I think that that’s been a consistent theme that I’ve seen where everyone just expects someone else to know what to do. And then as a parent, we must be like ‘come on, come on, come on!’ That’s my least favorite role in all of this.*
(Lauren)

### 3.3. Theme 3: Perioperative Support

All families reported that they themselves and their children felt fear, grief, stress, frustration, and pain. Roseanna shared “*I do remember that it was about the three-month mark. I was really sleep deprived and very depressed and that was the first time that has ever really happened to me. It was very, very difficult, very difficult.”* Five families reported that despite challenging experiences, they did not have access to counseling or receive mental health support. 


*I do remember one of the nurses asking if we wanted a psychiatrist. Which at the time, I wish I could have been clear-headed enough to say yeah, because it was a very, I mean, we thought we lost him, so it was very traumatic. But at the same time, I didn’t know that service was offered, and we’ve been frequent flyers at [the tertiary children’s hospital] for a long time. But yeah, it would have been useful, just maybe a couple days later when we had some sort of idea of what was going on.*
(April)

Lauren reflected on her experience of being in “*combat mode*”, trying to keep her daughter alive; she shared her appreciation for services such as Child Life and doctors who took the time to genuinely check in: “*I know we definitely had doctors that would look me in the eye in an appointment with Olivia and be like how are you doing? How are you coping? Which I always really appreciated.”*

Although most caregivers recalled receiving basic postoperative counsel (e.g., handouts or a conversation with healthcare practitioners), eight caregivers explained that they needed more guidance when they were discharged from the hospital. One caregiver described being taught how to use her son’s newly placed g-tube over Zoom^TM^, due to the surgery taking place during the COVID-19 pandemic, as “crazy”. April described the heavy responsibility placed on caregivers and the distress caused by having to make critical decisions on behalf of their children after receiving limited directives from the healthcare team: “*With us being sent home it was just a monitor, and you decide type of thing—here’s a prescription. Which has been stressful because we don’t know whether we’re over medicating or under medicating*.” Lauren said that


*We only found out that there was some sort of hip rehab maybe three years post-op… We don’t know why that was never offered to us. We got a pamphlet like last year, or something like that, and I’m like, we’re past this. There was a long time where we just didn’t know what we were supposed to do.*


Caregivers who reported feeling pleased by the guidance provided during the postoperative process often spoke to the following: receiving ongoing and thorough counseling on what to expect throughout the perioperative process; having supports in place for discharge, such as pediatric palliative care or home support nursing; receiving training for all caregivers involved in providing care instead of just the legal guardians; and having access to on-call healthcare support.


*I could text her [the neurosurgeon] pictures of what his incision looked like, and I don’t know, it was very comforting… Being able to page somebody at [the tertiary children’*
*s hospital] when you’ve discharged is amazing, especially with a kid like Jacob—we get help right away.*
(April)

In addition to hospital/community-based supports, eight caregivers revealed they valued connecting with peers whose children had experienced similar surgical interventions. Lauren shared that “*Surprisingly, you gain more information and more insight from parents who have gone through similar things than you get support from the hospital.*” Some families shared that they have found peer support and mentorship through online groups.


*We have a Facebook group of provincial parents of complex kids and it’s a pretty good support system. I have what I call my mentor mom here. Her son and my son are very similar, but her son is 21 so he’s kind of been through everything. I often lean on her for stuff.*
(April)

Two families revealed they did not have access to peer support and yearned for this type of connection. Lauren stated “I would have appreciated some sort of communication. I mean Zoom^TM^ wasn’t in vogue then, but I would have appreciated connection with other families.”

### 3.4. Theme 4: Changes to Caregivers’ Roles

Caregivers identified different occasions where they had to take on varying levels of responsibility in connecting and engaging with the healthcare system. Katie shared her concern regarding the time spent navigating the system, saying 


*That was the other thing too, just going back and forth to [the tertiary children’s hospital] all the time, getting adjusted to his chair… I don’t know how parents that work even part time, could keep up the number of appointments he had post-op, it was constant.*


Nicole shared her distress with navigating care:


*I have a kid with a G-tube now; who’s my point of contact for that? How do I get her supplies? Who do I call if I have a problem? And families must learn who to reach out to—you must learn connections, because nobody is going to do it for you.*


Seven out of twelve caregivers spoke to the need to become more versed in medicine to support their children. 


*I think every family that has a child like Taylor becomes a healthcare professional by choice. So yes, you’re a parent, but… like Taylor is a job to look after, and my husband is not medical, the grandparents aren’t medical, but just through caring for her they have medical skill and knowledge now.*
(Nicole)

Five caregivers revealed the need to speak up for their children in order to receive ideal care. Anne explained the need to advocate for increased monitoring and care of her child, while others advocated for better pain management and medication changes:


*He [the anesthetist] really liked his cocktail of heavy midazolam pre-op and she [Taylor] was like a zombie when I took her home. She typically wakes up super quick and pleasant and is ready to go, so I requested for the last surgery, which was tubes and her teeth, that there was no premedication with midazolam and she was fine again.*
(Nicole)

Nicole, who is trained as an ICU nurse, also expressed her desire to have a break and know that her child is in good hands, stating “*Well the thing is, I am her mom. I don’t want to be a nurse here; I have my own job. Take care of my kid; I shouldn’t have to be a nurse here*.”

## 4. Discussion

To the best of our knowledge, this is the first study to explore postoperative functional changes in children with SNI as reported by their primary caregivers. Eight out of twelve children in our study experienced severe complications defined by permanent functional changes and/or a recovery to baseline period that lasted beyond one year. There was no single domain of functional decline amongst these children, consistent with the observation by Jain and colleagues that GMFCS level V describes a heterogenous group of children that needs to be divided further to better predict postoperative risks and consequences [29]. By acknowledging these risks in advance, the health care team can be better prepared to conduct precise preoperative assessments, assist with the various transitions associated with surgery, and allow for a more stable recovery [30]. The postoperative changes in children identified herein are consistent with those reported by others, such as cardiac issues, respiratory decline and neurological decline [6]. Each child in this study experienced one or more issues.

Exactly why postoperative changes may occur is unknown. In the literature on adults, postoperative cognitive decline (PoCD) has been documented for over a century in elderly patients; however, there is a lack of consistent research methodologies, thus limiting the ability to compare these findings [7,8]. Historically, anesthesia exposure combined with pre-operative frailty has been proposed as the main cause for the onset of PoCD, yet the effects last far beyond the metabolic clearance of the anesthetics, and in Rasmussen’s work, there was no difference between patients receiving general anesthetic compared to those receiving regional anesthetic [7,10,11]. The limited research on PoCD in pediatric populations has led to questions regarding the safety of anesthesia administered to children during surgical procedures as well as questions related to its potential contribution to postoperative functional decline [31]. Berry et al. used the term ‘postoperative physiologic decline’ (PoPD) and designated PoPD by including the symptoms of cardiovascular, respiratory, and neurologic deterioration [6]. That study by Berry et al. found a one-in-four risk for PoPD in pediatric patients undergoing elective surgical procedures. The likelihood of PoPD increased by over a third in patients with complex chronic conditions (CCC) if they used eleven or more medications. A subset of children with cerebral palsy (CP) can be described as having SNI, and the recent literature indicates that children with CP often have reduced overall health, in turn making them vulnerable to perioperative complications from anesthesia and surgical intervention [32,33,34,35,36,37,38,39,40,41,42]. Although there is scarce research on the impacts of anesthesia for children with CP/SNI, the Buckon study indicates that having an active health issue is a risk factor for PoPD and reinforces the need for further research that considers the unique preoperative physical, cognitive, and communication factors of children with SNI [42]. Eleven of the twelve children in this study had both major and minor procedures. In these cases, there would have been a longer duration of anesthetic and at the same time more inflammation due to tissue disruption. Therefore, the mechanism(s) behind this phenomenon, whether due to anesthetic, inflammation, a combination of these, or some other factor, should be an active area of further research.

Furthermore, 11 out of 12 caregivers in this study reported that the clinical management of pain needed to be improved. They expressed a belief that the pain experience of their child was not fully understood and resulted in inappropriate pain management and further functional problems. Our findings are consistent with others that identify the need for objective pain measures that do not rely on self-report to capture the experience of pain [6,43]. Accurate pain assessment is challenging for a number of reasons, including improper and/or inadequate pain assessment tools being used; fewer assessments being completed; and a widely believed hypothesis that those with SNI process and experience pain differently [12,43].

Additionally, caregivers revealed the need for mental health services for children to be better integrated into the postoperative recovery experience. Caregivers in our study reported children with very limited communication (CFCS level V) experience significant, unaddressed psychological distress and postoperative stress symptoms after major surgeries. This study also revealed ways in which health providers can deliver more holistic support that acknowledges the psychological needs of caregivers and their families, such as improving access to mental health and peer support.

Many caregivers reported they felt their perspectives were undervalued and were not considered by healthcare providers, despite best knowing their child’s needs and their individualized means of communication [44]. There is a need for more opportunities within the hospital setting to hear the voices of caregivers, especially in relation to their child’s experience of pain. Sepehri et al. explains how disconnected systems particularly impact these families because children with SNI have conditions that are complex, life-long care in duration, and require a high number of care providers [45]. Postoperative care could further be improved with effective communication among care providers, improved clinical pain management, and by providing more guidance; this includes peer support and an on-call physician available to answer questions after discharge. A multidisciplinary team which includes rehabilitation experts can identify barriers for individuals with SNI that arise throughout the surgical process and provide support to optimize engagement in daily activities. Peri-operative administration of standardized assessments could enhance the identification of children at risk of postoperative functional decline and ensure appropriate postoperative home supports, equipment, and assistive technology are in place.

Finally, health providers can further reduce demands on caregivers to allow for improved quality of life and role balance by strengthening continuity of care between all providers, addressing unexpected events that arise upon discharge, and acknowledging the role of disruptions that can occur from significant life events, such as surgery.

Our study had several limitations which could affect the external validity of our findings. The participants we reached were all mothers or female caregivers and predominantly white, English-speaking, and middle-class. Our sample had a lack of diversity, limiting the translation of the findings to the male caregivers or that of individuals from different cultures or socioeconomic status.

## 5. Conclusions

Our results show that the phenomenon of postoperative changes in function in children with SNI lasting beyond 6 months is reported by caregivers. Given these findings, there is a need for more research that includes a more diverse sample of caregivers/children; integrates quantitative, standardized, and systematic measures with qualitative findings; and evaluates outcomes beyond one year postoperatively to more thoroughly identify the prevalence and impact of functional postoperative changes in children with SNI. Further research that explores ways to improve effective collaboration and communication among all providers and stakeholders is also needed, and one route may be through reimagining information management systems. Finally, studies aiming to understand the pathophysiological mechanisms underlying prolonged postoperative adverse functional changes in children with SNI are warranted.

## Figures and Tables

**Table 1 children-11-00319-t001:** Clinical and Functional Status of the Children.

Participant (Pseudonyms)	Child’s Diagnosis	Age	Sex	GMFCS Score	CFCS Score	Surgical Procedure	Year/Age of Procedure
Jean, mother to James	KSNTI-Gene, seizure disorder	16	M	V	V	Spinal surgery	2017 (age 14)
Roseanna, mother to Lucy	Rare genetic mutation, epileptic encephalopathy	15	F	V	V	Double full hip reconstructions Rod surgery	2010 (age 5) 2014 (age 13)
Beth/Holly, foster mother and paid caregiver to Tommy	Bilateral MCA infarction in utero with hydrocephaly and global developmental delay, quadriplegic CP with SNI, epilepsy, cortical visual impairment, gastroesophageal reflex disease, constipation, hip dysplasia	6	M	V	V	4 gastrointestinal surgeries	2018 (age 5)
Anne, mother to Graham	CP and seizure disorder	16	M	V	V	G-tube insertion 3 full hip reconstructions (2 surgeries on one side, 1 on the other) Rod surgery	2005 (age 1.5) 2007 (age 4) 2017 (age 14)
Katie, mother to Erik	Genetic condition, neurological decline without diagnosis, autism	16	M	V	V	Full hip reconstruction and tendon release on both legs	2014 (age 10)
April, mother to Jacob	Severe prematurity and CP	15	M	V	V	G-tube insertion De-rotational hip osteotomy Dental surgery Fixing malfunctioned shunt (two times) Removal of femur heads	2013 (age 5) 2015 (age 7) 2020 (age 12) 2021 (age 13) 2022 (age 14)
Lauren, mother to Olivia	Atypical CP with growth hormone deficiency and global developmental delay/ASD	6	F	V	V	G-tube insertion Bilateral open hip reduction	2016 (infant) 2017 (>1)
Mary, mother to Angelo	Dyskinetic CP secondary to neonatal, Kernicterus of unknown origin +/− underlying metabolic disorder and spasticity	7	M	V	V	G-tube insertion Cochlear implants bilaterally with dentistry (coordinated surgery)	2017 (age 1) 2020 (age 4)
Kate, mother to Nicholas	In utero stroke, epilepsy, cognitive and physical delays	17	M	V	V	Spinal correction	2018 (age 13)
Nicole, mother to Taylor	Treatment resistant epilepsy secondary to lissencephaly	14	F	V	V	G-tube insertion Ear tubes Ear tubes replaced (twice)	2012 (age 4) 2016 (age 8) 2019 (age 11) 2020 (age 12)
Nadine, mother to Charlie	Neonatal asphyxia	7	M	IV	IV	G-tube Double Varus Derotation Osteotomy (VDRO) Plates removed	2017 (age 2) 2020 (age 5) 2022 (age 7)
Amanda, mother to Arthur	CP, Enlarged tonsils, GERD, Global developmental delay, Hypothyroidism, Microcephaly, Reactive airway disease, Scoliosis, Seizure disorder, TUBG1 Gene mutation, VRE (vancomycin-resistant Enterococci), Visual impairment+	5	M	V	V	G-tube Hip reconstruction	2020 (age 2) 2023 (age 5)

**Table 2 children-11-00319-t002:** Number and type of post operative changes.

Postoperative Changes	Number of Participants
Cardiac arrest	1
Respiratory decline	6
Increase in seizures	4
Transient cognitive changes	2
Changes to communication	4
Increase in tone	2
Decreased sensation	1
Increase in restless leg syndrome/tremors in legs	2
Bladder function (UTIs, difficulty voiding)	1
Infections	8
Suck/swallow ability, weight loss	4
Increased gastro issues (reflux/gagging/vomiting/constipation)	4
Sleep quality	4
Decrease in sitting tolerance	3
Decrease in standing tolerance	2
Pressure sores/skin problems	2
Blood loss/blood transfusions	4
Increase in pain	10
Muscle atrophy during recovery leading to increase in other issues (i.e.; increase in spinal curvature or hip dislocation)	3
Worsened mental health	4
Changes to transfers	3
Clonus	1
Decreased posture/changes in alignment	2
Decreased ability to operate assistive tech	1
Changes in recreational participation (i.e., horseback riding, biking, swimming)	4

**Table 3 children-11-00319-t003:** Rating of Post Operative Recovery.

Time to Full Recovery	Number of Children
Uncomplicated Recovery (7–10 days in hospital and no unanticipated events)	0
Moderate Complications for Recovery (Resumed normal activities within 6 months)	2
Significant Complications for Recovery (Permanent changes and/or still recovering after 1 year)	10

## Data Availability

The data presented in this study are available on request from the corresponding author. The data are not publicly available due to privacy.

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
