# Peer review of "Acute and Persistent Postoperative Functional Decline in Children with Severe Neurological Impairment: A Qualitative, Exploratory Study"

_children, 2024, doi:10.3390/children11030319_

Round 1

Reviewer 1 Report

Comments and Suggestions for Authors

The title of the manuscript is adequate; I would only recommend including the type of the manuscript in the title.

Abstract is adequate

The introduction is adequate; only consider reading the “Instruction for authors” for adequate citation and reference.

Materials and methods.

Why were 12 individuals chosen? Could the authors explain the reason?

Was it a volunteer questionnaire?

How was the interview performed regarding scales? What were the specific questions used? Did the authors follow a specific questionnaire?

Results are adequate

Discussion

Could the authors comment regarding the external validity of this type of study?

Could the authors explain the difference between performing invasive and non-invasive surgeries? There is a significant difference between performing a GI surgery and an orthopedic surgery. A significant relationship with inflammatory pathways can be assumed to worsen neurological function.

Could the authors provide new ideas regarding future studies? Future studies without affecting external validity.

A conclusion is lacking. Could the authors write a conclusion section?

Author Response

Reviewer 1: The title of the manuscript is adequate; I would only recommend including the type of the manuscript in the title.

Response: We added a slight adjustment to the title to reflect the methodology.

Reviewer 1: Abstract is adequate.

Reviewer 1: The introduction is adequate; only consider reading the “Instruction for authors” for adequate citation and reference.

Response: We have revised the references.

Reviewer 1: Why were 12 individuals chosen? Could the authors explain the reason?

Response: We included 12 individuals as after this number of interviews we were not finding additional information that contributed further understanding to the themes/experiences of this exploratory study. We found data sufficiency. We have added a sentence in the methods to reflect this.

Reviewer 1: Was it a volunteer questionnaire?

Response: The participants agreed to be interviewed and to answer questions regarding their child’s mobility and communications to confirm their GMFCS and CFCS classifications.

Reviewer 1: How was the interview performed regarding scales? What were the specific questions used? Did the authors follow a specific questionnaire?

Response: The GMFCS and CFCS were both administered verbally to each participant to confirm inclusion criteria. We added a sentence to give a few examples of the questions we developed to explore the post-operative functional outcomes and impact on caregivers.

Reviewer 1: Could the authors comment regarding the external validity of this type of study?

Response: We have indicated how the limitations of our study could influence the external validity and added further recommendations to support more robust research regarding the long-term functional changes post-operatively in children with SNI. 

Reviewer 1: Could the authors explain the difference between performing invasive and non-invasive surgeries? There is a significant difference between performing a GI surgery and an orthopedic surgery. A significant relationship with inflammatory pathways can be assumed to worsen neurological function.

Response: We have added information to the discussion to highlight the potential interplay between surgery and inflammation and for the need for further mechanistic work to unravel the underlying processes that may be involved.

Reviewer 1: Could the authors provide new ideas regarding future studies? Future studies without affecting external validity.

Response: See earlier comments related to external validity.

Reviewer 1: A conclusion is lacking. Could the authors write a conclusion section?

Response: We have labeled and expanded the conclusion.

Reviewer 2 Report

Comments and Suggestions for Authors

Diversity and Inclusion: The participant demographic is limited predominantly to English-speaking, middle-class mothers, suggesting a need for a broader, more diverse sample that includes caregivers of different cultural, linguistic, and socioeconomic backgrounds to enhance the generalizability of findings.

Methodological Transparency: While the study uses thematic content analysis based on interviews, further clarity on the selection criteria for themes and the process of thematic analysis could strengthen the research's methodological rigor.

Longitudinal Follow-up: The study presents a snapshot of caregiver experiences post-surgery. Incorporating longitudinal follow-up could provide insights into the long-term trajectory of functional recovery or decline in children with SNI.

Quantitative Data Integration: While the qualitative approach offers depth, integrating quantitative measures of functional decline could provide a more comprehensive understanding of the impact of surgery on children with SNI.

Author Response

Reviewer 2: Diversity and Inclusion: The participant demographic is limited predominantly to English-speaking, middle-class mothers, suggesting a need for a broader, more diverse sample that includes caregivers of different cultural, linguistic, and socioeconomic backgrounds to enhance the generalizability of findings.

Response: We agree with this and have recommended further research be conducted to include a more diverse sample.

Reviewer 2: Methodological Transparency: While the study uses thematic content analysis based on interviews, further clarity on the selection criteria for themes and the process of thematic analysis could strengthen the research's methodological rigor.

Response: We have added two paragraphs with further details regarding the analysis.

Reviewer 2: Longitudinal Follow-up: The study presents a snapshot of caregiver experiences post-surgery. Incorporating longitudinal follow-up could provide insights into the long-term trajectory of functional recovery or decline in children with SNI.

Response: We agree and have added a recommendation related to long-term follow up in the conclusion.

Reviewer 2: Quantitative Data Integration: While the qualitative approach offers depth, integrating quantitative measures of functional decline could provide a more comprehensive understanding of the impact of surgery on children with SNI.

Response: We agree and have included this recommendation in the conclusion.